# Memory Efficient Tabular Foundation Models

Shuting Luo [1]   Monika Mikhail Kanaan [1]   Cameron Gordon [2]   Anna Leontjeva [1]   Simon Lucey [2]

## Abstract

Tabular Foundation Models, such as TabPFN, have received a large amount of recent attention due to their performance on in-context tabular machine learning tasks, which often exceeds classical baselines. However, practical deployment considerations of these models have received less attention. In this paper we investigate the memory requirements for these models. We demonstrate that employing model compression approaches can enable memory reductions of up to $7.6\times$ with similar levels of performance, reducing deployment requirements by nearly $87\%$. Our work provides insight to practitioners seeking efficient deployment of these models in practical settings.

## 1. Introduction

Tabular machine learning is used heavily in practical domains such as healthcare, finance, operations, and enterprise analytics, where models are often deployed under strict constraints on latency, cost, hardware availability, and data movement (Jiang et al., 2026; Shwartz-Ziv & Armon, 2021; Borisov et al., 2024). Recent tabular foundation models (TFMs), including the TabPFN and TabICL series, are attractive in these settings because they can make strong predictions in-context, reducing the need for task-specific training, hyperparameter tuning, and bespoke model-selection pipelines (Jiang et al., 2026; Hollmann et al., 2023; Qu et al., 2025; Grinsztajn et al., 2025). This makes them appealing for organizations that need to solve many small or medium-sized tabular prediction problems repeatedly.

However, practical adoption depends not only on predictive performance, but also on whether these models can be deployed and communicated efficiently (Fu et al., 2024). Enterprises must contend with memory movements, hardware constraints, and practical bandwidth considerations. These issues are well-known constraints in servicing large

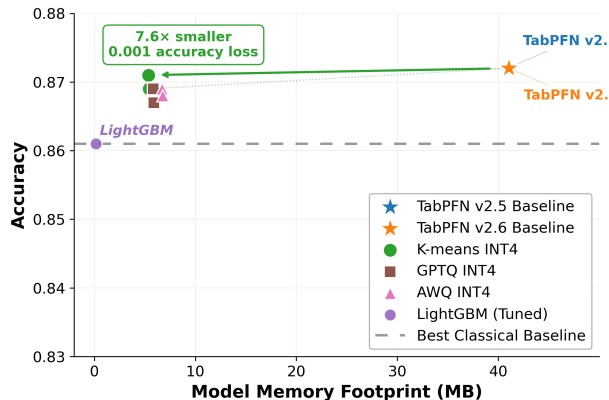

*Figure 1.* INT4 quantization makes Tabular Foundation Models substantially more memory-efficient while preserving their accuracy advantage. Quantized models have a $7.6\times$ lower memory footprint relative to full precision baselines, with negligible decline in accuracy, remaining above the strongest tuned classical baseline. This demonstrates the potential for quantized TFMs to achieve high predictive performance at a fraction of the deployment cost.

language models, where memory footprint directly affects deployment cost, latency, and the feasibility of local inference (Menghani, 2023; Jeyaraman, 2025). To address this, memory-efficient techniques such as model quantization are now ubiquitous in practical pipelines (Dettmers et al., 2023; Gholami et al., 2022).

These memory-based limitations also exist for TFMs, however to date most work has focused on reducing *context* constraints, due to architectural limitations and attention-based inference that grows quickly with sample and feature dimensions (Zabërgja et al., 2026; Grinsztajn et al., 2025; Qu et al., 2026; Zeng et al., 2024; Liu & Ye, 2025), leaving practical deployment considerations relatively unexplored.

This paper asks the question: **Can we reduce the deployment requirements for TFMs while retaining their predictive performance?** We evaluate whether existing quantization methods can be employed to reduce the memory footprint of the TabPFN family of models, using standard tabular benchmarks and classical baselines for context.

Our results indicate that quantization is a simple and practical route to memory-efficient TFMs. Across 30 datasets, we find that quantizing to INT4 precision reduces memory

[1]Commonwealth Bank of Australia, Australia [2]Australian Institute for Machine Learning, University of Adelaide, Adelaide, Australia. Correspondence to: Shuting Luo <aria.luo@cba.com.au>.

*The 2nd Foundation Models for Structured Data (FMSD) Workshop at the International Conference on Machine Learning*, 2026.

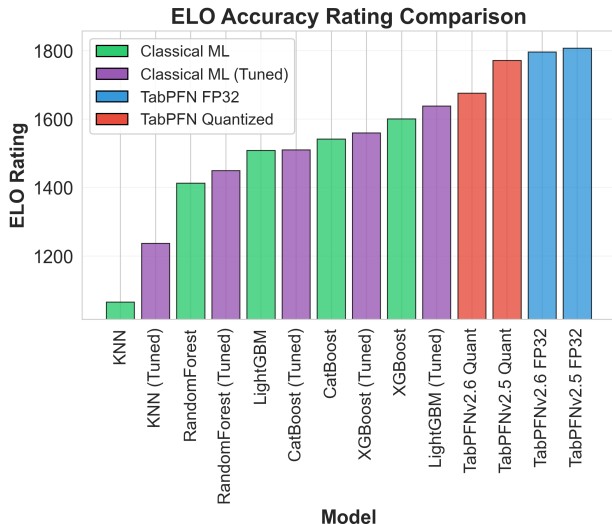

(a) ELO ratings from pairwise accuracy comparisons. Quantized TabPFN models remain close to their FP32 counterparts and rank above all tuned and untuned classical baselines.

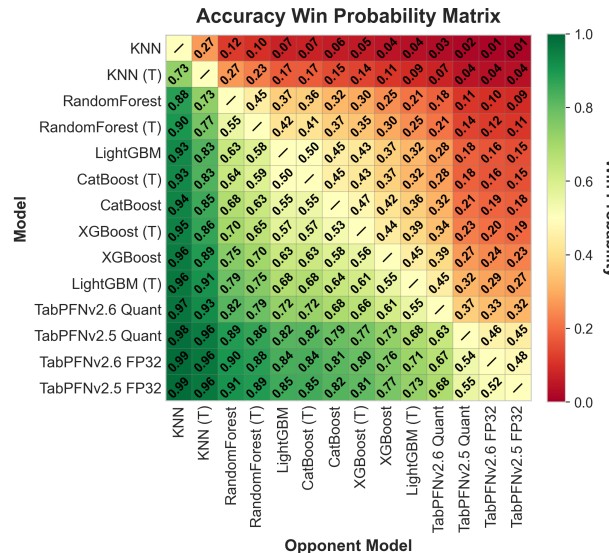

(b) Pairwise win probabilities on evaluated datasets. Quantized TabPFN models consistently beat classical baselines, with most losses concentrated against FP32 TabPFN variants.

Figure 2. Quantization preserves most of the accuracy gains of Tabular Foundation Models. Quantized TabPFN variants incur only a modest drop relative to FP32 models while maintaining a clear advantage over strong classical ML baselines.

footprint of TFMs while maintaining similar predictive performance to the full precision counterparts. These results show that deployment needs can be reduced by up to $7.6\times$ while maintaining performance comparable to tree-based models like CatBoost (Prokhorenkova et al., 2018).

In this paper we make the following contributions:

- We highlight the practical need for memory-efficient tabular machine learning models and demonstrate that large improvements can be obtained using standard quantization methods.

- We provide a systematic evaluation of these quantization approaches, and qualitative considerations for practical deployment.

We demonstrate our experiments on a range of standard tabular datasets using the TabPFNv2.5 and TabPFNv2.6 variants from the TabPFN family of models (Grinsztajn et al., 2025).

## 2. Related Literature

### 2.1. Tabular Foundation Models

Tabular Foundation Models are a class of pre-trained models that can be applied to tabular machine learning problems without additional training (Jiang et al., 2026). These models are trained to predict a posterior prediction ($y_{test}$) based on a tuple of in-context data ($x_{train}, y_{train}; x_{test}$) (Holl-

mann et al., 2023; Qu et al., 2025). The training of these models involves extensive pre-training on simulated data of causal relationships and are known as Prior Fitted Networks (PFNs) (Müller et al., 2022; Hollmann et al., 2023). Follow up work has sought to extend these models to more realistic tabular scenarios (e.g. missing data, categorical data, time-series), and to handle limitations in the feature and sample dimension (Hollmann et al., 2025; Grinsztajn et al., 2025; Qu et al., 2025; 2026; Hoo et al., 2025). The strong performance of these models relative to classical baselines has generated applied interest in a wide range of practical domains such as medical machine learning, financial forecasting, and a variety of industrial uses (Grinsztajn et al., 2025). The practical handling of large contexts has received a large amount of interest (Zeng et al., 2024; Zabërgja et al., 2026). However, memory requirements of weight deployment have received relatively little attention.

### 2.2. Quantization and Model Compression

Quantization is a mapping involving the discretization of a signal (Gersho & Gray, 1991). Within machine learning, the term is typically used to refer to *weight* or *activation* quantization, where the goal is to reduce memory by using lower precision values (Gholami et al., 2022).[1] We outline briefly

---

[1]Note: the term *quantization* is occasionally used to refer to discretization of features in tabular machine learning contexts. We use the term to refer to weight quantization in this paper reflecting our interest in memory efficiency in Tabular Foundation Models.

*Table 1.* Comparison of model performance and compression methods (30 Datasets).

| Model | Method | Accuracy | AUC | Memory (MB) | Compression Ratio |
|---|---|---|---|---|---|
| TabPFN-v2.5 | Baseline | 0.872 | 0.869 | 40.95 | 1.00x |
| | GPTQ INT4 | 0.869 | 0.866 | 5.81 | 7.04x |
| | KMeans INT4 | 0.871 | 0.867 | 5.36 | 7.65x |
| | AWQ INT4 | 0.869 | 0.866 | 6.70 | 6.12x |
| TabPFN-v2.6 | Baseline | 0.872 | 0.870 | 41.05 | 1.00x |
| | GPTQ INT4 | 0.867 | 0.865 | 5.89 | 6.97x |
| | KMeans INT4 | 0.869 | 0.868 | 5.43 | 7.56x |
| | AWQ INT4 | 0.868 | 0.866 | 6.77 | 6.06x |
| XGBoost | Baseline | 0.860 | 0.850 | 0.096 | – |
| | Tuned | 0.862 | 0.855 | 0.142 | – |
| CatBoost | Baseline | 0.858 | 0.851 | 0.073 | – |
| | Tuned | 0.861 | 0.856 | 0.099 | – |
| LightGBM | Baseline | 0.860 | 0.851 | 0.124 | – |
| | Tuned | 0.864 | 0.854 | 0.183 | – |
| KNN | Baseline | 0.798 | 0.704 | 0.203 | – |
| | Tuned | 0.828 | 0.755 | 0.220 | – |
| RandomForest | Baseline | 0.859 | 0.850 | 0.367 | – |
| | Tuned | 0.861 | 0.852 | 0.853 | – |

broad trends in the literature and key design decisions.

Early work includes binary neural networks (Rastegari et al., 2016; Hubara et al., 2016), the use of quantization aware-training, and integer quantization (Jacob et al., 2018). Classical results show that MSE optimal quantizers correspond closely to a clustering problem (Gersho & Gray, 1991), motivating codebook k-means quantizers (Han et al., 2016).

Recent advances in quantization have included the use of random rotations (Zandieh et al., 2026; Liu et al., 2025; Ashkboos et al., 2024); the use of singular value decomposition to reduce the effect of outlier values (Li* et al., 2025); activation-aware quantization (AWQ) (Lin et al., 2024); and dedicated libraries such as GPTQ and BitsandBytes (Dettmers et al., 2022a;b; 2023; Frantar et al., 2022). A large body of literature exists on applying quantization methods to reduce the memory requirements for modality specific use-cases such as large language model compression (Gholami et al., 2022). However, little work has been devoted to weight quantization within TFMs.

### 2.3. Memory Efficient Tabular Models

There is some work in reducing memory requirements for tabular machine learning models. For example, through the use of low-precision decision gradient boosted trees (Herrmann et al., 2025; Shi et al., 2022); and in applying integer quantization to enable efficient performance on FPGA-enabled edge devices (Alsharari et al., 2025). Separately, within TFMs there are some works on achieving high tabular performance with small MLP or distilled models such as TabM (Gorishniy et al., 2025) and TabDistill (Dissanayake

& Dutta, 2025). In addition, inference memory requirements have been reduced by compressing context (Zabërgja et al., 2026), or through hypernetworks (Mueller et al., 2025).

## 3. Method

### 3.1. Quantization Methods

We evaluate three INT4 post-training quantization approaches indicative of recent trends in model compression: K-Means Quantization (Han et al., 2016), AWQ (Lin et al., 2024), and GPTQ (Frantar et al., 2022). We focus on post-training quantization (PTQ) rather than quantization-aware training, as this can be conducted on a pre-trained model and does not require access to the original pipeline. For GPTQ and AWQ, originally developed for autoregressive language models, we adapt their calibration procedures to TabPFN's in-context learning setting by treating the $(x_{train}, y_{train}; x_{test})$ tuple as the calibration input. For K-Means clustering quantization, we implement an INT4 bit-packing scheme to reduce model artifact size on disk.

### 3.2. Experimental Setup

We evaluate on a subset of 30 datasets from the OpenML library (Vanschoren et al., 2013), selected to match feature and sample limitations of TabPFN. A full list of datasets is provided in Appendix A.1. We test our three quantization methods on two TabPFN variants: TabPFNv2.5, and TabPFNv2.6 (Grinsztajn et al., 2025). Memory is evaluated following BZIP2 compression, replicating a practical deployment pipeline (Seward, 1996). We benchmark against

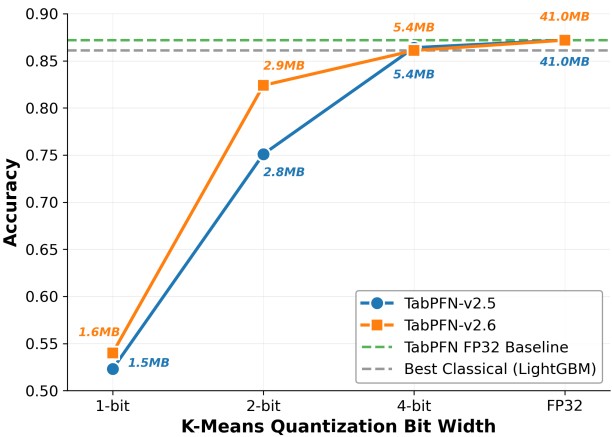

*Figure 3.* Effect of K-means quantization bit width on TabPFN accuracy and model size. Accuracy improves sharply from 1-bit to 2-bit quantization and largely saturates by 4-bit quantization, where both TabPFN-v2.5 and TabPFN-v2.6 match or approach full precision performance while requiring only 5.4 MB. Further increasing to FP32 yields negligible accuracy gains but increases model size substantially to 41.0 MB, indicating that 4-bit quantization provides the best accuracy-memory trade-off.

standard tabular machine learning methods: XGBoost (Chen & Guestrin, 2016), CatBoost (Prokhorenkova et al., 2018), LightGBM (Ke et al., 2017), KNN (Fix & Hodges Jr, 1951; Cover & Hart, 1967), and Random Forest (Breiman, 2001), evaluated in both default-configuration and tuned variants. Hyperparameter details are provided in Appendix A.2.

## 4. Results

Table 1 and Figure 1 present quantization performance across TabPFNv2.5 and TabPFNv2.6. All three INT4 quantization methods achieve substantial compression (6.0-7.65×) with minimal accuracy degradation ($\leq$ 0.005). Among quantization methods, K-means INT4 achieves the highest compression ratio (7.65× for TabPFNv2.5, and 7.56× for TabPFNv2.6) while incurring the smallest accuracy loss (0.001), outperforming both GPTQ and AWQ across both model versions. Critically, all quantized variants maintain accuracy above the best classical baseline (LightGBM tuned: 0.864), demonstrating that compression preserves TabPFN's performance advantage over traditional methods.

Several further observations support these results. First, a bit-width ablation (Table 2) confirms INT4 as the practical optimum: 2-bit quantization recovers only ~90% of baseline accuracy and 1-bit collapses to near-random performance, while INT4 retains over 99%. Second, Figure 2 shows that the quantized models retain a consistent dataset-level advantage over classical baselines, with a marginal relative ELO loss relative to full precision versions. Third, performance variance across the 30 datasets remains essen-

tially unchanged between baseline and quantized variants (see Appendix Table 3), indicating that quantization does not introduce dataset-dependent instability. Finally, while deployable memory is substantially reduced, inference-time memory and runtime are largely unchanged (see Appendix Table 4). This is consistent with TabPFN's inference cost being dominated by attention activations rather than weights, motivating activation-level compression as future work.

## 5. Conclusion and Limitations

In this work, we investigated the memory efficiency of TFMs, with a particular focus on quantization-based model compression. Our results demonstrate that substantial reductions in deployment memory are achievable with quantization without compromising predictive performance, making these models more practical for use within industry.

**Limitations** Our study focuses exclusively on post-training quantization, which may be suboptimal relative to quantization-aware training, which may produce higher performance at lower-precision with access to the training pipeline (Rastegari et al., 2016; Han et al., 2016; Gholami et al., 2022). In addition, our evaluation considers only a subset of available quantization techniques, leaving more advanced approaches such as TurboQuant (Zandieh et al., 2026) and other emerging methods unexplored. As is standard in quantization for memory deployment, our method requires dequantization prior to inference and so does not enable run-time memory improvements (Gholami et al., 2022). We also do not incorporate kernel-level or hardware-aware optimizations, which may enable FPGA/edge-device implementations. Finally, our method retains the architectural limitations of the foundation models tested which constrain the maximum context window (Grinsztajn et al., 2025).

**Future Work** Several promising directions exist for extending this work. Beyond quantization, complementary forms of model compression including pruning (Han et al., 2016), low-rank adaptation (LoRA) (Hu et al., 2022), and knowledge distillation (Hinton et al., 2015), may yield further gains in memory and computational efficiency. Improving inference-time memory efficiency represents another valuable avenue, for example by combining memory-efficient attention mechanisms or context-handling strategies with a quantized approach. Exploring quantization-aware training may further improve efficiency while preserving low-precision predictive performance (Rastegari et al., 2016; Han et al., 2016). Activation quantization strategies, which enable inference in low-precision are a further avenue (Gholami et al., 2022). Finally, integrating hardware-aware optimizations and extending TFM models to physical edge devices remains an exciting potential future direction for a range of practical memory-bound applications.

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

# A. Additional Details

## A.1. Datasets

All datasets were evaluated by standardised train:test split as provided by Open ML dataset tasks. Details of the datasets used are provided below:

| Dataset | Samples | Features | Classes | Description |
| --- | --- | --- | --- | --- |
| Amazon employee access | 32,769 | 9 | 2 | Amazon employee access prediction |
| anneal | 898 | 38 | 5 | Steel annealing process classification |
| Bank Customer Churn | 10,000 | 10 | 2 | Bank customer churn prediction |
| Bank marketing | 45211 | 13 | 2 | Bank marketing classification |
| blood-transfusion-service-center | 748 | 4 | 2 | Blood transfusion donor behavior |
| churn | 5,000 | 19 | 2 | Telecom customer churn prediction |
| coil2000 insurance policies | 9,822 | 85 | 2 | Insurance policy purchase prediction |
| credit card clients default | 30,000 | 23 | 2 | |
| credit-g | 1,000 | 20 | 2 | German credit risk assessment |
| diabetes | 768 | 8 | 2 | Pima Indians diabetes diagnosis |
| E-CommereShippingData | 10,999 | 10 | 2 | |
| Fitness Club | 1,500 | 6 | 2 | Fitness club membership churn prediction |
| hazelnut-spread-contaminant-detection | 2,400 | 30 | 2 | Food contamination detection |
| heloc | 10,459 | 23 | 2 | |
| HR Analytics Job Change of Data Scientist | 19,158 | 12 | 2 | |
| in vehicle coupon recommendation | 12,684 | 24 | 2 | |
| Is-this-a-good-customer | 1,723 | 13 | 2 | Customer quality classification |
| jm1 | 10,885 | 21 | 2 | |
| Marketing Campaign | 2,240 | 25 | 2 | Marketing campaign response prediction |
| maternal health risk | 1,014 | 6 | 3 | Maternal health risk classification |
| MIC | 1,699 | 111 | 8 | Text/molecular classification |
| NATICUSdroid | 7,491 | 86 | 2 | Android malware detection |
| online shoppers intention | 12,330 | 17 | 2 | |
| polish companies bankruptcy | 5,910 | 64 | 2 | Polish company bankruptcy prediction |
| qsar-biodeg | 1,054 | 41 | 2 | Chemical biodegradability prediction |
| seismic-bumps | 2,584 | 15 | 2 | Mining seismic hazard prediction |
| splice | 3,190 | 60 | 3 | DNA splice junction classification |
| students dropout and academic success | 4,424 | 36 | 3 | Student academic outcome prediction |
| taiwanese bankruptcy prediction | 6,819 | 94 | 2 | Taiwan company bankruptcy prediction |
| website phishing | 1,353 | 9 | 3 | Phishing website detection |

## A.2. Classical Comparisons

**Baseline Methods**    Baseline models were run for comparison using fixed default configurations across multiple independent datasets. The results provided are the mean across all datasets for each model. Chosen parameters are included for reference:

```
XGBoost
    n_estimators: 100
    max_depth: 6
    learning_rate:0.1
    random_state: 42
    n_jobs: -1
    eval_metric: 'logloss'
    objective: 'binary:logistic' (binary) or 'multi:softmax' (multi-class)
    use_label_encoder: False

CatBoost
    iterations: 200
    depth: 6
    learning_rate: 0.1
    random_state: 42
    verbose: False
    thread_counts: -1
    loss_function: 'Logloss' (binary) or 'Multiclass' (multi-class)

LightGBM
    n_estimators: 100
    max_depth: 6
    learning_rate:0.1
    random_state: 42
    n_jobs: -1
    verbose: -1
    force_col_wise: True

RandomForest
    n_estimators: 100
    max_depth: 10
    random_state: 42
    n_jobs: -1
    verbose: 0

KNN
    n_neighbors: 5
    n_jobs: -1
```

**Tuning Methods**    `RandomisedSearchCV` was used with stratified 3-fold cross validation (20 random hyperparameter configurations per model) using the training dataset and reporting the optimized model performance on the test data. Multiple parameter grid search spaces were used covering regularization, tree depth, learning rates, and other model-specific parameters. Details of the parameter search grids are included below:

```
XGBoost (9 parameters)
    max_depth: [3, 5, 7, 10]
    learning_rate: [0.01, 0.05, 0.1, 0.2]
    n_estimators: [50, 100, 200, 300]
    min_child_weight: [1, 3, 5]
    subsample: [0.6, 0.8, 1.0]
```

```
        colsample_bytree: [0.6, 0.8, 1.0]
        gamma: [0. 0.1, 0.2]
        reg_alpha: [0, 0.1, 0.5, 1.0]
        reg_lambda: [0.5, 1.0, 2.0]

    CatBoost (7 parameters)
        iterations: [50, 100, 200, 300]
        depth: [4, 6, 8, 10]
        learning_rate: [0.01, 0.05, 0.1, 0.2]
        l2_leaf_reg: [1, 3, 5, 7]
        border_count: [32, 64, 128]
        bagging_temperature: [0, 0.5, 1.0]
        random_strength: [0.5, 1.0, 2.0]

    LightGBM (9 parameters)
        num_leaves: [15, 31, 63, 127]
        max_depth: [3, 5, 7, 10, -1]
        learning_rate: [0.01, 0.05, 0.1, 0.2]
        n_estimators: [50, 100, 200, 300]
        min_child_samples: [10, 20, 30, None]
        subsample: [0.6, 0.8, 1.0]
        colsample_bytree: [0.6, 0.8, 1.0]
        reg_alpha: [0, 0.1, 0.5, 1.0]
        reg_lambda: [0, 0.1, 0.5, 1.0]

    RandomForest (7 parameters)
        n_estimators: [50, 100, 200, 300]
        max_depth: [5, 10, 20, 30, None]
        min_samples_split: [2, 5, 10]
        min_samples_leaf: [1, 2, 4]
        max_features: ['sqrt', 'log2', 0.5, 0.7]
        bootstrap: [True, False]
        criterion: ['gini', 'entropy']

    KNN (5 parameters)
        n_neighbours: [3, 5, 7, 9, 11, 15, 21]
        weights: ['uniform', 'distance']
        metric: ['euclidean', 'manhattan', 'minkowski']
        p: [1, 2, 3]
        algorithm: ['ball_tree', 'kd_tree', 'brute']
```

## B. Hardware Details

Experiments were conducted on an AWS SageMaker `ml.g4dn.12xlarge` instance ($4\times$ NVIDIA T4 GPUs with 16 GB VRAM each, 48 vCPUs and 192 GB of shared RAM). Four worker processes were run in parallel; each worker was pinned to one GPU and a dedicated set of 12 vCPUs to prevent inter-worker contention. RAM was shared across workers rather than strictly partitioned, as physical memory cannot be isolated in the same way as compute resources. Per-process peak resident set size (RSS) during inference is reported in Table 4.

# C. Ablations

## C.1. K-means N-bit ablation study

*Table 2.* K-means N-bit ablation study

| Model | Bit Width | Accuracy | ROC-AUC | Model Size (MB) | Compression Ratio |
|---|---|---|---|---|---|
| | Baseline FP32 | 0.865 | 0.910 | 40.95 | 1.00x |
| TabPFN-v2.5 | 1-bit | 0.523 | 0.535 | 1.52 | 26.94x |
| | 2-bit | 0.751 | 0.863 | 2.80 | 14.63x |
| | 4-bit | 0.864 | 0.906 | 5.36 | 7.64x |
| | Baseline FP32 | 0.864 | 0.906 | 41.05 | 1.00x |
| TabPFN-v2.6 | 1-bit | 0.540 | 0.558 | 1.59 | 25.82x |
| | 2-bit | 0.824 | 0.875 | 2.87 | 14.30x |
| | 4-bit | 0.861 | 0.905 | 5.43 | 7.56x |

*Note: Results computed on 5 representative datasets and may differ slightly from 30-dataset averages in Table 1 due to dataset sampling.*

4-bit quantization achieves the optimal trade-off between compression and performance, recovering over 99% of baseline accuracy at 7.64× compression. 2-bit quantization offers a viable alternative at 14-15× compression with ∼90% accuracy recovery. 1-bit quantization is impractical for production use due to severe accuracy degradation (∼40%).

## C.2. Performance stability across datasets

*Table 3.* Performance stability study across 30 classification datasets

| Model | Method | Accuracy | ROC-AUC | Balanced Acc |
|---|---|---|---|---|
| | awq INT4 | 0.869±0.087 | 0.866±0.090 | 0.712±0.175 |
| TabPFN-v2.5 | baseline FP32 | 0.872±0.086 | 0.869±0.089 | 0.720±0.172 |
| | gptq INT4 | 0.869±0.087 | 0.866±0.090 | 0.710±0.174 |
| | kmeans INT4 | 0.871±0.085 | 0.867±0.090 | 0.718±0.171 |
| | awq INT4 | 0.868±0.089 | 0.866±0.091 | 0.714±0.170 |
| TabPFN-v2.6 | baseline FP32 | 0.872±0.086 | 0.870±0.090 | 0.722±0.170 |
| | gptq INT4 | 0.867±0.088 | 0.865±0.091 | 0.714±0.171 |
| | kmeans INT4 | 0.869±0.087 | 0.868±0.090 | 0.721±0.167 |

*Note: Results are averaged across 30 datasets. Standard deviation represents performance variation across datasets.*

Standard deviation across datasets remains consistent between quantized and baseline variants, indicating that quantization does not introduce additional performance variance across different data distributions.

## C.3. Inference memory and speeds

*Table 4.* Inference memory and speed averaged across 30 OpenML datasets.

| Model | Method | Peak RSS (MB) | Inference Mem* (MB) | $\Delta$ vs Baseline | Speedup vs FP32 |
|---|---|---|---|---|---|
| TabPFN-v2.5 | FP32 | 5,893 | 4,737 | - | 1.00x |
| | GPTQ INT4 | 5,953 | 4,798 | +61 (+1.3%) | 1.10x |
| | KMeans INT4 | 5,867 | 4,712 | -25 (-0.5%) | 0.99x |
| | AWQ INT4 | 5,923 | 4,767 | +30 (+0.6%) | 1.05x |
| TabPFN-v2.6 | FP32 | 5,717 | 4,561 | - | 1.00x |
| | GPTQ INT4 | 5,791 | 4,634 | +73 (+1.6%) | 1.00x |
| | KMeans INT4 | 5,727 | 4,571 | +10 (+0.2%) | 0.91x |
| | AWQ INT4 | 5,775 | 4,619 | +57 (+1.3%) | 1.00x |

*\*Inference Mem = Peak RSS - Baseline RSS (process overhead removed).*

Table 4 presents an analysis of inference efficiency. Our primary contribution focuses on weight compression to address deployment bottlenecks (model distribution and storage), achieving up to 7× compression. As an additional analysis, we measure inference memory, which remains largely unchanged ($< 2\%$ variation, $\sim$4.7 GB), as expected since memory usage is dominated by activations. This observation motivates future work on activation quantization. Inference speed remains stable (±10%), indicating no performance degradation from weight quantization.

