# OpenReview forum: "Memory Efficient Tabular Foundation Models"
_ICML.cc/2026/Workshop/FMSD — FMSD @ ICML 2026 Poster_

### Official Review · Reviewer_pPr1 · 2026-05-13
**Strong motivation with convincing results**

**Rating:** 8
**Confidence:** 4

**Review:**

**Summary**
Although memory limitations regarding maximum context length for Tabular foundation models have already been considered in the literature, model size remains underexplored. The authors propose shrinking tabular foundation models (specifically TabPFNv2.5 and TabPFNv2.6) using four different compression methods to make them deployable for practitioners. The results show that TabPFN is compressible by a factor of around 7.6 times while still being able to beat classical baselines such as CatBoost.

**Strengths**
The motivation is very clear, and the topic is relevant for the workshop. The text is easy to follow, and decent baselines were chosen.

**Areas for Improvement**
In some parts, the rationale behind decisions in the method is not 100% clear to the reader. Reproducibility via a dedicated GitHub repository would further strengthen the contribution.

**Detailed Comments**
Parts that could be improved without too much extra space:
- In some parts, the authors claim that accuracy drops a bit (still over catboost as in Figure 1), while in different parts, the authors claim that the accuracy drops are "negligible". As the improvements compared to other TabFMs such as TabICL are more and more marginal in terms of accuracy, such performance drops, although seeming low, may be important. Please weaken such phrases in the text.
- It would be good to also evaluate your results on the TabArena or TALENT benchmark, as this would strengthen the narrative and further support the claims made.
- A repository for reproducibility or open-sourcing the compressed models would be useful.
- It would be a good insight to not only plot the Accuracy change with respect to the compression, but also the change in AUC. This could be done in the appendix.

Some minor errors that I came across while reading:
- Line 153/154: "Tnere".
- Line 202: "x" used twice.
- Table 3: TabPFNv2.5 and TabPFNv2.6 4-bit have ROC-AUC far beyond the baseline. This may be an indicator that those 5 datasets really are not that representative.
- Under table 5: Please do not refer to a table with "this table".
- B. Hardware Details: Why is the total RAM 186GB, and then only 40GB per worker?

**Justification of Score**
Overall, the paper shows a promising step towards compressing tabular foundation models such as TabPFN to save memory. The narrative is clear, and the results support the claims made. Although more comprehensive experiments could be made, this paper is a strong contribution.

---

### Official Review · Reviewer_EpFU · 2026-05-20
**Useful but thin: off-the-shelf INT4 quantization applied to TabPFN, with honest results that quietly undercut the headline deployment claim**

**Rating:** 6
**Confidence:** 4

**Review:**

Strengths:

1. Addresses a genuinely underexplored angle (deployment/storage memory for TFMs, vs. the well-studied context-memory problem) with a clean empirical study across 30 datasets, 2 TFM variants, and 3 quantization methods plus tuned classical baselines.

2. Honest, useful ablations: the bit-width sweep (1/2/4-bit) cleanly justifies INT4 as the sweet spot, and the inference-memory table transparently shows that weight quantization does not reduce runtime memory (dominated by activations) - preventing over-claiming.

Weaknesses:

1. he contribution is almost entirely an application of off-the-shelf PTQ methods to TabPFN with no methodological novelty; the practical payoff is also limited because inference memory and runtime are essentially unchanged (the actual deployment bottleneck for TFMs), making "87% deployment reduction" a storage/transfer claim that the framing somewhat oversells.

2.Several reporting issues: only accuracy/AUC reported (no calibration, NLL, or significance tests despite 30 datasets); the Table 3 bit-width ablation is on 5 datasets while Table 1 is on 30 (inconsistent and noted only in a footnote);

---

### Official Review · Reviewer_tPLX · 2026-05-22
**Practical Storage Gains but Minimal Runtime Efficiency for TabPFN**

**Rating:** 6
**Confidence:** 4

**Review:**

### **Summary**

This paper investigates the application of post-training weight quantization (PTQ) to Tabular Foundation Models (TFMs), specifically focusing on the TabPFNv2.5 and v2.6 variants. The authors evaluate three standard INT4 (among others levels) quantization methods, K-Means, GPTQ, and AWQ across 30 OpenML datasets. The results demonstrate that models can be compressed by up to 7.6x (reducing storage from approximately 41.0 MB to 5.4 MB) with a negligible accuracy loss of roughly 0.001. Despite this compression, the quantized TFMs maintain a performance advantage over strong, tuned classical baselines like CatBoost.

*Crucial Note:* It is vital to state clearly that this approach does not make the model fit onto smaller or memory-constrained hardware. While weight storage is significantly reduced , the inference activations remain entirely untouched at full precision.

### **Strengths**
- **Practical Empirical Benchmarking:** The paper provides a clear, documented roadmap for practitioners looking to minimize the storage and distribution footprint of TabPFN using off-the-shelf methods.

- **Methodological Breadth:** Evaluation across 30 datasets with both default and tuned classical lightweight baselines (XGBoost, CatBoost, LightGBM, Random Forest, and KNN) ensures the comparative results are robust.

- **Clear Optimization Curve:** The n-bit ablation study (Table 3) effectively justifies INT4 as the optimal trade-off point for TabPFN, demonstrating that 1-bit and 2-bit quantization lead to a severe performance collapse.

- **High-Quality Presentation:** The figures (particularly Figures 1 and 2) and tables are clean and effectively communicate the trade-offs between *storage footprint* and predictive performance.

### **Areas for Improvement**
- **Misleading Framing of "Deployment Requirements":** The Abstract and Introduction claim to reduce "deployment requirements by nearly 87%". However, Table 5 (and later mentioned in the future work and lightly acknowledged in the Experiments) reveals that Inference Memory (Peak RSS) changes by less than 2% because the memory bottleneck is overwhelmingly dominated by attention activations rather than static weights. The unclear framing hints a reduction in runtime hardware constraints that does not exist, which deserved a point int the limitations section.

- **Limited Technical Novelty:** The work applies existing, well-known quantization strategies (GPTQ, AWQ, K-Means) to a specific model family without introducing new compression architectures or handling the structural challenges of Tabular Foundation Models.

- **Incomplete Quantization Scope:** By leaving activations at full precision, the authors avoid the primary technical hurdle of TFM deployment. The reported accuracy stability is highly expected given that the core intermediate computations remain at full precision. The paper more or less demonstrates that the weights are robust to noise, but it does not achieve true execution efficiency or proof that we can deploy these models to less capable hardware.

- **Lack of Target Hardware Profiling:** Following the last point, the authors motivate the work by citing "hardware availability" and constraints. However, the experiments were conducted on a powerful AWS SageMaker ml.g4dn.12xlarge instance with an Nvidia T4 GPU. To truly prove the value of this compression, the paper would benefit from latency and memory profiling on actual constrained hardware (e.g., CPU-only environments or edge devices) to see if the smaller disk size provides tangible benefits beyond just faster download speeds.

### **Detailed Comments**
- **Table 5 to the Main Text:** The data regarding Inference Memory and Speeds  is arguably the most important context in the paper for practitioners. Relegating it to the Appendix obscures the reality that the runtime memory footprint (approx. 4.7 GB ) remains unchanged.

- **Clarify "Deployment" in the Introduction:** Please revise the claims in the Abstract and Introduction  to explicitly state that the proposed quantization reduces "model storage and distribution requirements" rather than general "deployment requirements." This will prevent readers from falsely assuming the model can now be run on 1GB-RAM edge devices.

- *Question for the Authors:* Given that Prior-Fitted Networks (PFNs) are pre-trained on simulated data, did you consider Quantization-Aware Training (QAT) by simulating quantization noise directly in the prior? While you state PTQ was chosen for simplicity, mentioning QAT as a specific, mathematically aligned future direction for TabPFN would strengthen the discussion.

### **Justification of Score**
**Rating: Marginally above acceptance threshold**

This paper provides a solid, clean empirical baseline that will be useful for the workshop community as a quick citation for weight compression in Tabular Foundation Models. The authors rigorously evaluate standard quantization methods across 30 datasets and against strong baselines, successfully identifying INT4 as the optimal trade-off point.
However, the technical novelty is low, and the framing of "deployment efficiency"  is misleading. Because the activations remain at full precision, the actual runtime RAM requirements  are fundamentally untouched, meaning the paper solves a storage bottleneck rather than an execution one. Despite this limitation, it is a well-executed empirical study that fits the scope of the FMSD workshop.